# Probiotics as a Possible Strategy for the Prevention and Treatment of Allergies. A Narrative Review

**DOI:** 10.3390/foods10040701

**Published:** 2021-03-25

**Authors:** Aroa Lopez-Santamarina, Esther Gonzalez Gonzalez, Alexandre Lamas, Alicia del Carmen Mondragon, Patricia Regal, Jose Manuel Miranda

**Affiliations:** Laboratorio de Higiene Inspección y Control de Alimentos, Departamento de Química Analítica, Nutrición y Bromatología, Universidad de Santiago de Compostela, 27002 Lugo, Spain; aroa.lopez.santamarina@usc.es (A.L.-S.); esther.gonzalez.gonzalez@rai.usc.es (E.G.G.); alexandre.lamas@usc.es (A.L.); aliciamondragon@yahoo.com (A.d.C.M.); patricia.regal@usc.es (P.R.)

**Keywords:** allergic disease, probiotic, atopic dermatitis, rhinitis, gut microbiota

## Abstract

Allergies are an increasing global public health concern, especially for children and people living in urban environments. Allergies impair the quality of life of those who suffer from them, and for this reason, alternatives for the treatment of allergic diseases or reduction in their symptoms are being sought. The main objective of this study was to compile the studies carried out on probiotics as a possible therapy for allergies. The most studied allergies on which probiotics have been shown to have a beneficial effect are rhinitis, asthma, and atopic dermatitis. Most studies have studied the administration of *Lactobacillus* and *Bifidobacterium* spp. in children and have shown beneficial effects, such as a reduction in hyperreactivity and inflammation caused by allergens and a decrease in cytokine release, among other beneficial effects. In the case of children, no clear beneficial effects were found in several studies, and the potential risk from the use of some opportunistic bacteria, such as probiotics, seems controversial. In the studies that reported beneficial results, these effects were found to make allergy symptoms less aggressive, thus reducing morbidity in allergy sufferers. The different effects of the same probiotic bacteria on different patients seem to reinforce the idea that the efficacy of probiotics is dependent on the microbial species or strain, its derived metabolites and byproducts, and the gut microbiota eubiosis of the patient. This study is relevant in the context of allergic diseases, as it provides a broader understanding of new alternatives for the treatment of allergies, both in children, who are the main sufferers, and adults, showing that probiotics, in some cases, reduce the symptoms and severity of such diseases.

## 1. Introduction

An allergy is defined as a hypersensitivity reaction caused by an immunological response to a specific antigen, known as an allergen. Allergens act on innate immune cells. Repeated contact with an allergen triggers the activation of mast cells and basophils and the release of allergic mediators, resulting in symptoms ranging from sneezing and itchy rashes to severe shortness of breath and anaphylaxis [1,2,3]. Currently, the list of allergic diseases published by the WHO includes asthma, rhinitis, conjunctivitis, rhinosinusitis, anaphylaxis, atopic eczema, hives, and angioedema, as well as secondary reactions caused by drugs, foods, or insects [4].

Today, approximately 1 billion people worldwide suffer from allergies, and these numbers are estimated to increase to 4 billion in the next 30–40 years [1]. In recent years, the prevalence of allergic diseases has increased greatly, to the point that 30–40% of the world’s population now suffers from one or more allergic diseases. It is important to note that these diseases affect all age groups, but they are more frequently reported in childhood [5].

While no determinant risk factors have been identified, it is possible that environmental factors, such as cigarette smoking, air pollution, and exposure to allergens explain the observed changes in the prevalence of allergic diseases [6]. A common explanation for the increased prevalence of allergic diseases is the “hygiene hypothesis”, first proposed in 1989 by Strachan [7], in which it is stated that a lack of exposure in childhood to infectious agents, symbiotic microorganisms, and parasites increases the child’s susceptibility to these diseases later in life [6]. Decreased microbial exposure due to improved hygiene, changes in diet, or increased use of antibiotics may be determinants of an increased prevalence of allergies [7,8,9]. Additionally, there is a lot of evidence suggesting that living in rural areas in childhood has a protective effect against allergic disease, such as atopy, allergic rhinitis, and asthma [10,11]. This may be related to environment-derived factors, such as contact with bacterial endotoxins [12], contact with animals [13], or intake of raw or lightly processed milk [13,14]. In recent decades, the migration from traditional farming to the urban environment, increase in processed food intake, lack of contact with animals, and excessive hygiene has been related to the increase in the incidence of allergic diseases [15,16].

Due to the increase in allergies, their prevention and treatment have become a global public health priority. For this reason, new alternatives are being sought, one of which could be the use of probiotics to prevent these diseases and reduce their symptoms [5]. “Probiotic” means “for life”, and probiotics are defined as “Live microorganisms that, when being administered in appropriate doses, confer a benefit to the health of the host” [5]. The use of probiotics to treat or prevent specific diseases is a branch of microbiology known as “therapeutic microbiology” [17,18,19,20]. Probiotics exert multiple health effects, such as immunomodulatory agents and activators of host defense pathways, which reduce morbidity [21]. In the context of allergic disease, the probiotics’ microbiome is essential for the development of host immune responses. Effective probiotics must be resistant to bile salts, gastric enzymes, and a low pH, and they cannot cause mucosal inflammation or infection [22].

The population of microorganisms that inhabits a determined ecological niche of healthy individuals is termed the intestinal microbiota. Its colonization begins before birth and matures slowly, until it reaches the adult state at around 3 years of age. The human gut microbiota establishes a symbiotic relationship with the host and plays a very important role in human health, because dysbiosis in the gut microbiota often occurs in the presence of disease [17,20]. The intestinal microbiota constitute a key factor in the development of an adequate immune response, because the contact between gut bacterial antigens and the part of the immune system associated with the intestine represents an important component of the human immune system [17]. The microbiota play a fundamental role in the development of the human immune system, especially in the first years of life. Its main role is to intervene in the homeostasis and immunity of the intestine [17].

The main objective of this study is to provide a literature review of the effects of the administration of probiotics in the prevention and treatment of allergic diseases. This study is relevant, because it broadens the knowledge of new alternatives for the treatment of these diseases in both children and adults, reduction in the symptoms associated with allergies, and prevention of the occurrence of allergen-caused acute adverse episodes.

## 2. Methodology

A narrative literature search was conducted up to 10 February 2021 for all the available literature in the following databases: Web of Science, PubMed, and Scopus. A combination of the following search terms was applied: “probiotics” and “allergic diseases”; “gut microbiota” and “allergic diseases”. The search terms were adjusted to the specific databases and consisted of a combination of free-text searches. The selection of articles will be limited to studies published in English and Spanish, with no restrictions on the year of publication, although the most prominent articles are those published after 2015. A total of 115 articles were selected and included in the review. The following data on the study characteristics were extracted from the included records: author and year of publication, type of study, probiotics used, dosage and time of administration, type of allergy, and the conclusions of each study. The authors reviewed the titles and the abstracts. If the abstracts reported the use of diaries that contained narrative elements, full texts were read, and if the pre-established eligibility criteria were met, they were included in the review. A narrative approach containing summary tables and graphs will facilitate the synthesis of the included studies.

## 3. Results and Discussion

### 3.1. Characteristics of Allergic Diseases and Most Common Allergic Diseases

In allergic diseases, there is a disturbance in the balance between T helper (Th)1 and Th2 lymphocytes in favor of Th2 lymphocytes. Th1 lymphocytes are key to infection, activating macrophages to defend the body primarily against intracellular microbes. Th2 lymphocytes, on the other hand, activate eosinophils and mast cells and induce the production of IgE, which is responsible for allergies. These allergies are caused by the response of an inappropriate immune response of Th2 lymphocytes to different antigens, including environmental or food antigens. The activation of this response leads to the secretion of interleukins (IL)-4, IL-5, and IL-13 and an allergen-specific IgE production, which drives allergic inflammation [1,10]. Interferon (INF)-γ inhibits Th1 activity by inducing these cytokine responses, thus maintaining an allergic phenotype [23].

The prevalence of allergic diseases has doubled in the industrialized parts of the world in the last 25 years [6]. These diseases mainly affect children and young people, but as they age, the severity and complexity of these diseases increase, resulting in an adverse impact on their quality life and high costs to the health care system [10,24]. In general terms, approximately 200–250 million people suffer from food allergies, one-tenth of the population suffers from drug allergies, and 400 million suffer from rhinitis [24]. Worldwide, 300 million people suffer from asthma, and this incidence is expected to increase by 100 million in 2025 [25]. Food allergies are recognized as the most common immune disorders [26], and they are considered a world health risk, particularly in developed countries [27]. The major risk factors for the development of food allergies are related to genetics, the environment, and immune tolerance failure. Additionally, gut microbiota composition and activity have an important role in immunological development, and thus, gut microbiota eubiosis is recognized a key factor in preventing food allergies [26,28].

One variable on which these allergic diseases depend is sex. Due to their characteristics and specific patterns in women, women are more susceptible than men to allergies due to the ovarian hormones and hyperreactivity of the airways, which may occur during the menstrual cycle or pregnancy [29,30]. Previous studies showed that a significant percentage of women with asthma suffer worse symptoms during the perimenstrual phase [31,32].

Allergic rhinitis (AR) is foremost among the most common allergic diseases. It affects between 10% and 20% of the total population and is therefore the most prevalent chronic non-communicable disease in the world [33]. However, the prevalence of this disease is probably underestimated because many patients do not recognize rhinitis as a disease and therefore do not seek medical advice [34]. The most important allergens that trigger AR are pollens. AR results from immunoglobulin (IgE)-mediated inflammation of the nasal mucosa, which is characterized by pruritus, sneezing, rhinorrhea, and nasal congestion. In addition, AR is a potential risk factor for asthma [24], another common allergic disease and chronic inflammatory disorder of the airways, which currently affects some 300 million people worldwide [24,35]. Asthma is a heterogeneous disease, in which inflammation of the airways occurs and respiratory symptoms, such as wheezing, chest tightness, and coughing, along with shortness of breath, which varies in duration and intensity [36].

Urticaria is a disorder with various underlying causes. It is estimated that approximately 25% of people experience at least one episode of urticaria in their lifetime, but only 3% will develop chronic urticaria [37]. It is characterized by the appearance of hives, lasting between 1 and 24 h, and/or angioedema, which can last up to 72 h [24]. The activation of mast cells located superficially in the skin leads to urticaria, while mast cells in the dermis are involved in angioedema. Histamine is the main mediator in urticaria and in many cases of angioedema [24,38].

The most common chronic inflammatory skin disease is atopic dermatitis (AD). The incidence of AD has increased 2- to 3-fold in recent years in industrialized countries [39,40]. In the last 30 years, the prevalence of this disease has increased to 10–20% in children and 1–3% in adults. This chronic inflammatory skin disease is also known as atopic eczema, is very pruritic, and is characterized by erythema and oedema. In addition, AD usually occurs during infancy and childhood [41]. AD may have a mixed IgE-mediated and non-IgE-mediated mechanism [10]. Atopy manifests as allergic rhinitis, bronchial asthma, atopic dermatitis, or even food allergies and is defined as the development of an immediate hypersensitivity reaction to environmental and food antigens due to a genetic predisposition [24,35]. A variant of AD is allergic contact dermatitis (ACD), which manifests due to a delayed hypersensitivity reaction and is mediated by T-cells in the skin. In this case, the allergens, which are called “haptens”, are of a low molecular weight and bind to proteins in the skin, where they become antigenic [42].

Food-origin allergies have a high morbidity, which affects the sufferer’s quality of life; involve high costs; and, in the case of anaphylaxis, can lead to death. Worldwide, 220–250 million people may suffer from food-origin allergies [43]. The foods most frequently implicated in pediatric patients are cow’s milk and eggs. Moreover, milk or egg sensitization in children is associated with an increased risk of developing dust mite allergy and even asthma. In many cases, food allergies are mediated by IgE [10]. In adulthood, the most common foods implicated in allergies are legumes, nuts, fruits, and crustaceans [43].

Anaphylaxis is the most serious allergic disease and can be fatal. The onset can be within minutes or even hours and usually affects different body systems. The triggers of anaphylaxis vary with age and geography, and the prevalence is at least 1% [44]. Anaphylaxis is usually IgE-mediated and can be due to food, insect venom, drugs, or latex [10].

### 3.2. The Immune System and Probiotics

The human intestinal microbiota constitutes a complex ecosystem that includes, in addition to bacteria, fungi, Archaea, viruses, and protozoa. The concentration of bacteria increases from the stomach the duodenum, but it is in the large intestine where it rises up to 10^11^–10^12^ UCF/g [45]. It has been estimated that there are at least 1800 genera and between 15,000 and 36,000 species of bacteria in the large intestine [17]. Firmicutes and Bacteroidetes are the main bacterial phyla, followed by Actinobacteria, Proteobacteria, and Verrucomicrobia. In addition, fungi and protozoa make up approximately 1% of the species in our gut microbiota [45,46].

“Probiotic” means “for life”, and probiotics are currently used to refer to bacteria that have beneficial effects on human and animal health [17]. In 2001, the Food and Agriculture Organization of the United Nations (FAO)/World Health Organization (WHO) defined them as “live microorganisms which, when administered in adequate amounts, confer a health benefit on the host” [47]. This definition was corrected in 2014 by the International Scientific Association for Probiotics and Prebiotics, and it now reads “microorganisms for which there is scientific evidence of safety and efficacy” and excludes “live cultures associated with fermented foods for which there is no evidence of a health benefit” [48].

Probiotics contribute various beneficial effects on human health and are used to treat different infectious and non-infectious diseases. Various effects have been seen in the form of protection against infections, decreases in irritable bowel symptoms, the inhibition of *Helicobacter pylori* growth and viral infections [49], the prevention of cancer, decreases in gut inflammatory response, and the prevention and/or treatment of allergies, which is the focus of this review [5].

It is believed that the administration of beneficial microorganisms may be the key to improving health and susceptibility to disease, as human organisms and the gut microbiota establish a symbiotic relationship important for the maintenance of human health [17]. It is likely that these microbiota organisms have evolved along with our immune system, thus promoting immune tolerance. This includes the induction of regulatory T cells (Treg) and the control of Th2 and Th1 balance, which may prevent the development of allergic and autoimmune diseases [1].

Probiotics can stimulate the immune system compounds secreted or present in the cellular barrier. Therefore, probiotic use improves the body’s defenses by triggering an immune response, according to the pathological state, and promoting a balance between pro- and anti-inflammatory cytokines that are secreted by activated immune cells. Probiotics hence act as a non-specific adjuvant to the innate immune response [50]. There is evidence that probiotics promote the production of some cytokines, including IL-10, transforming growth factor (TGF)-β, IL-12, and INF-γ, which regulate the immune response and reduce allergic inflammation [10].

Many studies have shown that the gut microbiota and probiotic intake can support maturation of the immune system during the first years of life due to different physiological and metabolic reactions in the host. The most promising probiotics in terms of immune system development are those belonging to the genera *Lactobacillus* and *Bifidobacterium* [6]. The effects of probiotics are dose- and strain-dependent [51,52] and may be influenced by age-specific functions, such as the maturity of the host’s intestinal barrier. It was suggested that the perinatal period may represent a window of opportunity for effective probiotic intervention in allergic diseases [53]. In the first stage of life, the microbiota is still developing, so the administration of probiotics leads to a proper microbial colonization and results in a greater effectiveness in the prevention and treatment of different diseases [17]. The immune effects of probiotics, which are primarily mediated through the innate immune system, include the promotion of epithelial integrity, intestinal permeability, and mucus production through the production of anti-inflammatory cytokines and tolerogenic CD103+ dendritic cells, in addition to the promotion of the differentiation and proliferation of regulatory T cells, the inhibition of the Th2 cell response, and an increased release of IgA from plasma cells [54,55]. The therapeutic potential of probiotics in allergic diseases is mediated by various mechanisms of action, such as the modulation of the immune response, competitive inhibition of invasive flora in the gut, modification of pathogenic toxins and host products, and an increased epithelial barrier function [41].

*Lactobacillus* reduces proinflammatory responses by regulating nuclear factor kappa B (NF-κB) signaling. Probiotics also promote the maturation of dendritic cells (DCs) into anti-inflammatory cytokines, such as IL-10. In addition, human monocyte-derived DCs can release IL-10 when treated with probiotics, thus triggering the differentiation and survival of Tregs [23]. *Bifidobacterium animalis* and *Bifidobacterium longum* induce the release of interferon gamma (IFN-γ) and tumor necrosis factor alpha (TNF-α) by DCs, while only *Bifidobacterium bifidum* can activate Th17 cells through the release of IL-17. Different studies have indicated that probiotics can modulate the Th1/Th2 balance and, consequently, prevent inflammatory diseases, such as allergies [23].

### 3.3. Use of Probiotics in Allergic Diseases

The increased prevalence of allergies, such as eczema, may be due to alterations in the gut microbiota in early life, such as a reduced abundance of the Proteobacteria phylum and the general *Bifidobacterium*, *Akkermansia*, *Faecalibacterium,* and *Lachnospira* [52,56]. For example, an association has been described between gut colonization with *Clostridium difficile* and allergies in children. Various studies have shown that children with atopic dermatitis have a less diverse gut microbiota [56,57] and lower *Bifidobacterium* levels [57] than healthy children. In allergic children, higher counts of *Staphylococcus aureus* and Enterobacteriaceae and lower *Bifidobacterium* counts were found than in healthy children [58]. In addition, a lower colonization of enterococci during the first month of life and of *Bifidobacterium* during the first year of life have been found in infants with allergies, compared to non-allergic infants [58].

It has been proposed that probiotics could potentially restore intestinal homeostasis and prevent or alleviate allergies by interacting with the intestinal immune cells [51]. Moreover, symbiotics, which are a synergistic combination of probiotics and prebiotics, have proven beneficial for different allergic conditions, for example, by reducing asthma-like symptoms and the use of asthma medications [59].

Probiotics are useful for the modulation of allergic diseases, as they stimulate the levels of IgA in the mucosa and the T and B cells of the immune system [23]. The results of different studies examining the influence of probiotics on allergic diseases have been contradictory, and as a result, few practical recommendations as to the use of probiotics in allergic diseases have been established [60]. It is important to note that, due to multiple factors, the results vary between studies (Figure 1). The impact of probiotics on humans is highly variable between individuals. For example, a recent study found that individuals with a healthier gut microbiota composition responded in a better way than other individuals to the administration of probiotics [61]. This result is logical, because the action of probiotics depends on several factors, including colonization resistance, acid and short chain fatty acid production, the competitive exclusion of pathogens, and bacteriocin production, which can influence probiotic persistence [62,63]. Zmora et al. [64] demonstrated that persistent personalized gut mucosal resistance to commercial probiotics is associated with unique host and microbiome features. For example, a probiotic’s colonization can be affected by an under-representation of specific carbohydrate-utilization genes in the gut, thus preventing the complex metabolization of polysaccharides by the probiotic bacteria [62]. Thus, the efficacy of probiotics is dependent on both the microbial species or strain and its derived metabolites and byproducts (commonly named as postbiotics), as well as the number of probiotic cells and type of probiotic carrier [27].

Previous studies have found that supplementation with various probiotic preparations can balance the intestinal microbiota, regulate the immune system, and reduce allergies. Further, these studies have explored new strains of probiotics, probiotic genomic characteristics, and the in vivo mechanisms of action, formulation processes, and safety of probiotics [65,66]. Currently, studies are attempting to identify protective factors that regulate the immune system and allow for tolerance of different allergens in a specific way. One of the most studied probiotics is *Lactobacillus* GG, which is frequently studied in connection with the management of AD, although the results have been quite inconsistent [8]. Further, *Lactobacillus* were able to reduce allergic symptoms and to reduce morbidity in children [67].

Taniuchi et al. [68] demonstrated that a *Bifidobacterium* species was effective in the treatment of cow’s milk hypersensitivity in infants with atopic dermatitis. In addition, Enomoto et al. [69] demonstrated that prenatal and postnatal supplementation with two species of *Bifidobacterium* reduced the risk of developing eczema and atopic dermatitis in infants. These differences disappeared at 10 months of age, highlighting that the early stage microbiota is particularly important for regulating allergies in children.

Two systematic reviews and meta-analyses show that probiotic supplementation before and after birth can prevent allergic diseases, particularly eczema [70,71]. On the other hand, when probiotics are administered only after birth, they may not be effective in preventing allergic diseases [71,72].

Many animal studies, mainly in mice, in which different probiotics in isolation or mixtures of them were used to treat or prevent allergic diseases, show an effect. Obviously, experimental animals differ widely from humans in key aspects, such as the gut microbiota composition, immune function, or metabolism [45]. Thus, extrapolating the results obtained from animal models to humans may not be valid and should always be considered as preliminary or limited.

The probiotics most often used in these studies are the *Lactobacillus* and *Bifidobacterium* genera and, to a lesser extent, *Enterococcus* [73]. The use of *Enterococcus* as probiotics presents some intrinsic risks that makes them poorly suited to use as probiotics, especially the *E. faecium* and *E. faecalis* species. As opportunistic pathogens, both enterococci possess an array of virulence factors and antibiotic resistance determinants, which render them controversial. Even in avirulent enterococci, when they reach the intestinal tract, they can interact with endogenous enterococci and other commensals that could lead to bi-directional genetic exchange [65,66].

In terms of allergies, the most studied are those related to the airways, such as asthma, but also atopic dermatitis [74,75] and food allergies [6,76]. The beneficial effects of probiotics on allergies include a reduction in hyperreactivity and inflammation due to the presence of allergens, a decrease in interleukins and eosinophils, and a reduction in TNF and INF, etc.

In addition to the animal studies, there have also been many studies with humans, especially children (Table 1). As in the studies with mice, in this case, the probiotics most often used were of the genera *Bifidobacterium* and *Lactobacillus*, or mixtures of them. To a lesser extent, *Propionibacterium* [77] or *Escherichia coli* and *Enterococcus faecalis* [8] were used, and these probiotics were administered in doses of 10^7^ to 10^10^ colony-forming units (CFU)/day. Today, the consensus statements consider a number of viable cells of 1 × 10^9^ CFU/day in food and food supplements as the minimum number to be effective [48]. To a lesser extent, food and respiratory allergies have been studied. The beneficial effects found in these studies were a reduction in inflammatory cells, decrease in interleukins, reduction in TNF and INF, and, above all, reduction in symptoms and improvement in the quality of life of people suffering from these allergies. Researchers, such as Simpson et al. [78] and Loo et al. [79], followed up their studies for several years to observe the effects of probiotics over time. Studies with a follow-up of 5 years or longer show that the greatest protective effect of probiotics against AD occurs in early childhood and that this effect is less likely to be sustained until school age [78].

As shown in (Table 1), most studies employed *Lactobacillus* or *Bifidobacterium* spp. as the probiotics. However, other bacterial species were also employed, such as *Propionibacterium freudereichii* ssp *shermanii* JS [77], *Escherichia coli* DSM 17252, and *Enterococcus faecalis* DSM 1644 [8] or *Streptococcus thermophilus* TH-4 [52]. The use of opportunistic pathogens, such as *Escherichia coli* or *Enterococcus faecalis* in infants, is controversial due to the abovementioned risks related to genetic exchange of virulence factors and antibiotic resistance determinants [65,66]. In all but three cases [67,86,92], the probiotics were employed to alleviate symptoms or prevent skin allergic diseases.

The effects obtained are controversial, because in several cases [41,52,85,86,87], no significant differences were found between the probiotic- and placebo-treated groups. One of the studies that did not find any significant change was a multicenter study trial with more than 1000 infants. For the cases in which significant effects were found, in most cases, the beneficial effects were slight, and in two studies, the beneficial effects were limited to specific child subgroups [8,83].

The results found regarding probiotic usage in adults and in both mother–child pairs with the aim of treating or prevent allergic diseases can be found in (Table 2).

As shown in (Table 2), most studies employed probiotics against skin allergic diseases, but they were also used against allergic rhinitis or asthma [55,89,94,99], Japanese cedar pollinosis [58], food allergies [94], and rhinoconjunctivitis [106]. All studies employed *Lactobacillus* or *Bifidobacterium* spp. as the probiotic strains, with only one exception, where mixed *Propionibacterium freudereichii* ssp *shermanii* JS were employed [94]. Unlike in the case of infants, in the case of adults or mother-infant pairs, the administration of probiotics had positive effects, except in two cases [98,104]. In these two cases, where *Lactobacillus rhamnosus* GG were employed in pregnant women, no beneficial effects were found.

It is important to note that probiotics are considered safe due to their widespread use in food and dairy products, even more so than a hundred years ago [23]. Studies such as systematic reviews and meta-analyses show that the use of probiotics is safe in preterm infants [108]. However, several risks can be associated with probiotic therapy in the clinical field and in vulnerable target groups, such as pregnant women, infants, and immunodeficient subjects [23]. Medical studies have shown that, although uncommon, the use of probiotics may have some side effects, such as the spread of inappropriate resistance genes in intestinal microbial populations, virulence factors in probiotic microbial strains, translocation to tissues and blood, an inflammatory response, and infections [27,109]. Commercial probiotics have also been found to contain live microorganisms not listed on their labels, which can be dangerous to an individual [110,111]. In fact, a premature infant has been reported to have died from fulminant gastrointestinal mucormycotic due to fungal contamination of a probiotic supplement used. Therefore, it is also important to use probiotics with some caution [110,111].

In addition, it should be noted that some of the studies collected have not clearly shown the effects of probiotics in treating allergies [52,84,87,97]. The studies that did not find beneficial effects on humans included those that applied *Lactobacillus* [82,95], mixes with *Bifidobacterium* [87], or *Streptococcus* [52] to preterm infants [52], children [52,84], and pregnant women [97]. The only link between these studies is that in all cases, probiotics were used for the treatment or prevention of atopic dermatitis, so it seems that in this pathology, the use of probiotics seems only minorly effective for this purpose.

Probiotics can be taken as pharmaceutical preparations or as functional foods, generating wider acceptance among consumers, mainly as fermented foods [49]. As the intake of functional foods shows a better consumer acceptance than pharmacological presentations, there is potential for foods that are consumed regularly when they are converted to functional foods [49]. It is true that interest is increasingly focused on the potential applications of food-derived products that can be used as tools to prevent or delay the onset of a health problem, in this case allergies. This has given rise to nutraceuticals, which are defined as “the phytocomplex if derived from a food of plant origin, and as the pool of secondary metabolites if derived from a food of animal origin, concentrated and administered in the most appropriate pharmaceutical form”. This does not stop here, but nanotechnology has enabled progress to be made towards nanonutraceuticals, which have advantages due to their remarkable properties and versatility. Alternatively, nanonization strategies for probiotics and the usefulness of nanoprobiotics in releasing encapsulated bacteria and enhancing their absorption in the gastrointestinal tract are being explored, which may be of great interest in the treatment of allergies [112].

There are also studies showing that the intake of fermented foods, such as fermented yeast, galactooligosaccharides, or fructooligosaccharides, can improve the symptoms of some allergies, such as asthma or eczema, by reducing oxidative stress and inflammation [113,114,115]. This suggests that the combined administration of probiotics and prebiotics could be useful for alleviating the symptoms of allergic patients.

## 4. Limitations, Conclusions, and Future Trends

Some of the studies have limitations, e.g., the reduced sample size, the randomization process, and the small number of bacterial groups used. In some cases, the probiotic dosage employed is below the minimum consensus (10^9^ UFC/day) for the appropriate use of probiotics. Thus, the results obtained in the studies that did not reach the minimum consensus limit of the appropriate use of probiotics are of doubtful validity.

“Probiotic” means “for life”, and probiotics are defined as “live microorganisms that, when being administered in appropriate doses, confer a benefit to the health of the host”, and although the use of most probiotics is generally safe, the use of some bacterial opportunistic pathogens, such as *Enterococcus* or *Escherichia*, is controversial and should be avoided, especially in vulnerable collectives, because they can interact with endogenous commensal pathogens with bi-directional genetic exchange. Previous studies have shown that, although uncommon, the use of probiotics may have some side effects.

This review has collected different published studies that show the beneficial effects of using probiotics to treat allergic diseases. No clear beneficial effects were found in children, because in several studies, no significant differences were found between probiotic- and placebo-treated groups. Even considering that probiotics do not cure allergies, their administration can reduce the morbidity and the duration of allergy symptoms. An alternative to the administration of probiotics in supplement form would be to consume them in functional foods, which are foods produced not only for their nutritional characteristics, but also to fulfil a specific function, such as improving health and/or reducing the risk of disease, as these would be more acceptable to consumers than pharmacological presentations and easier to consume. Another increasingly emerging way would be in the form of nanoprobiotics.

While many studies have been conducted, more clinical trials in humans are needed to observe the true effects of probiotic consumption on allergic diseases, but this review gives an idea of the great benefits that are likely to result from the use of probiotics in the treatment of allergies and the administration of probiotics alongside other treatments, thus reducing the morbidity of people living with allergies. Additionally, the efficacy of probiotics is dependent also on factors such as the composition and activity of the subject’s intestinal microbiota and the derived metabolites and byproducts synthesized by the probiotic species. Thus, prior knowledge of the state of the subject’s intestinal microbiota prior to the use of the probiotic agent would be advisable to improve its effectiveness.

## Figures and Tables

**Figure 1 foods-10-00701-f001:**
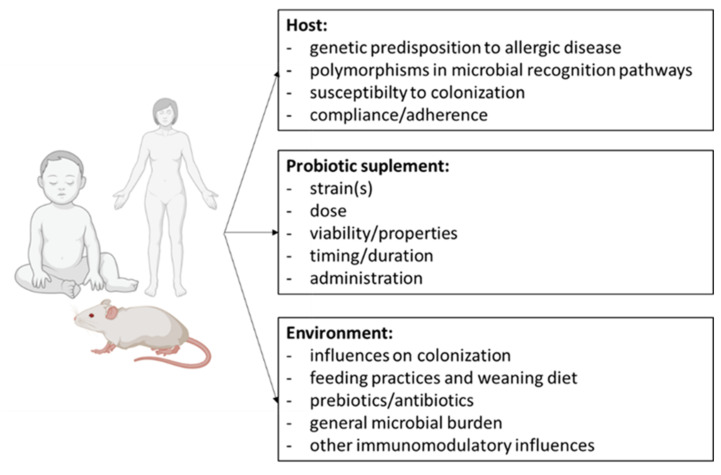
Factors that could explain the varied effects of probiotics uncovered by different studies.

**Table 1 foods-10-00701-t001:** Use of probiotics to treat or prevent allergic diseases in infants.

Type of Study	Probiotic	Dosage and Time of Exposure	Allergic Disease	Main Findings	Reference
Randomized, double-blind study with 27 infants with atopic disease	*Bifidobacterium lactis* Bb-12 and *Lactobacillus rhamnosus* GG (LGG)	Oral administration of 3 × 10^8^ CFU of LGG and 10^9^ CFU of *B. lactis* for 4 weeks	Atopic eczema	After 2 months, a significant improvement in the skin condition occurred in the probiotic group. The Scoring Atopic Dermatitis (SCORAD) and concentration of soluble CD4+ decreased in the probiotic groups	[80]
Double-blind, placebo-controlled, crossover study with 58 children	*Lactobacillus rhamnosus* 19070-2 and *Lactobacillus reuteri* DSM 122460	A dose of 10^10^ CFU twice daily for 6 weeks	Atopic dermatitis	The duration of eczema decreased during probiotic administration. The treatment response was more pronounced in allergic patients, and the SCORAD score decreased	[81]
Randomized, double-blind, placebo-controlled study with 230 infants with suspected cow’s milk allergy (CMA)	LGG, *L. rhamnosus* LC705, *B. breve* Bb99 and *Propionibacterium freudenreichii* ssp *shermanii* JS	Oral administration of LGG (5 × 10^9^ CFU) or a mixture of LGG (5 × 10^9^ CFU), L. *rhamnosus* LC705 (5 × 10^9^), *B. breve* Bb99 (2 × 10^8^), and *Propionibacterium freudenreichii* spp. *shermanii* JS (2 × 10^9^) twice daily for 4 weeks.	Atopic dermatitis related to cow’s milk allergy	Treatment with LGG may alleviate symptoms of atopic eczema and/or dermatitis syndrome in IgE-sensitized infants	[77]
Randomized, double-blind, placebo-controlled study with 56 children	*Lactobacillus fermentum* VRI-033 PCC	1 × 10^9^ CFU twice daily for 8 weeks	Atopic dermatitis	A reduction in the SCORAD index was seen in the probiotic-treated group. At the end of the study, more children treated with this probiotic had milder atopic dermatitis	[82]
Randomized, placebo-controlled study with 59 children with AD	*L. rhamnosus* and *B.* *lactis*	A dose of 2 × 10^10^ CFU daily for 12 weeks	Atopic dermatitis	A combination of *L. rhamnosus* and *B. lactis* improved atopic dermatitis only in food-sensitized children	[83]
Randomized, double-blind, placebo-controlled study with 50 infants	*L. rhamnosus* Lrh and LGG	Administering 1, 5, 25 and 125 mL of cow’s milk formula at 30 min intervals (5 × 10^9^ CFU/mL formula)	Atopic dermatitis related to cow’s milk allergy	No clear effects were seen on the SCORAD, sensitization, inflammatory parameters, or cytokine production	[84]
Randomized trial with newborns of 231 women with allergies	*Lactobacillus acidophilus* LAVRI-A1	3 × 10^9^ CFU/day for the first 6 months of life	Atopic dermatitis	Atopic dermatitis rates were similar in the probiotic and placebo groups. At 12 months, the rate of sensitization was significantly higher in the probiotic group	[85]
Randomized, double-blind, placebo-controlled study with 193 infants diagnosed with cow’s milk allergy (CMA)	*Lactobacillus casei* CRL431 and *Bifidobacterium lactis* BB-12	10^7^ CFU/g for each of the probiotic bacteria for 6 months	Cow’s milk allergy (CMA)	Supplementation with *Lactobacillus* and *Bifidobacterium* in an extensively hydrolyzed formula did not accelerate cow’s milk tolerance in infants with CMA	[86]
Double-blind, placebo-controlled, randomized trial with 253 infants	*B. longum* BL999 and *L. rhamnosus* LPR	Oral supplementation with 1 × 10^7^ CFU/g/day of *B. longum* and 2 × 10^7^ CFU/g/day of *L. rhamnosus* for the first 6 months	Atopy and eczema	No significant effect on the prevention of eczema or allergen sensitization in the first year of life	[87]
Double-blind, placebo-controlled, randomized trial with 179 infants	*Lactobacillus* F19	1 × 10^8^ CFU/day for 4–6 months	Eczema	The cumulative incidence of eczema at 13 months was lower in the probiotic group. At 13 months, the INF-y/IL-4 ratio was higher in the probiotic group. No differences in serum concentrations of IgE	[88]
Randomized, double-blind trial of children from 415 mothers	LGG, *B. animalis* ssp. *lactis* Bb-12 and *L. acidophilus* La-5	Milk contained 5 × 10^10^ CFU/day of *L. rhamnosus* and Bb-12. 5 × 10^9^ CFU of *L. acidophilus* for 4 months, from 36 weeks of gestation to 3 months postnatally	Atopic dermatitis and asthma	In the probiotic group, the cumulative incidence of atopic dermatitis was reduced but there was no effect on sensitization	[89]
Randomized, double-blind study with 39 infants with AD	LGG	A daily intake of 3.4 × 10^9^ CFU for 3 months	Atopic dermatitis	The proportions of IgA- and IgM-secreting cells decreased in the probiotic group, and the proportions of CD191+ and CD27+ B cells increased	[90]
Randomized, placebo-controlled trial with 606 newborns	*Escherichia coli* DSM 17252 and *Enterococcus faecalis* DSM 1644	Oral bacteria lysate containing heat-killed nonpathogenic 1.5–4.5 × 10^7^ bacteria/mL (3 × 0.7 mL/day)	Atopic dermatitis	A significant effect was observed in a subgroup of the probiotic group with single heredity for atopy, which was most pronounced for infants with atopic fathers	[8]
Double-blind, placebo-controlled, randomized parallel study with 100 children	Mixture of *L. casei, L. rhamnosus, L. plantarum,* and *B. lactis*	Oral administration at 2 × 10^9^ CFU in each strain, twice daily for 6 weeks	Atopic dermatitis	The probiotic mixture did not suppress the growth of other strains, but no differences in clinical improvement were seen between the treated and placebo groups	[41]
Double-blind, prospective, randomized, placebo-controlled study with 2020 children	*L. paracasei* GMNL-133 and/or *L. fermentum* GM090	2 × 10^9^ CFU/day of L. *paracasei*, *L. plantarum*, or 4 × 10^9^ CFU of a mixture for 3 months	Atopic dermatitis	Children given either probiotics alone or a mixture of both showed a decrease in the severity of atopic dermatitis scores. Lower test scores were also recorded for people with skin diseases. IgE, TNF-α, and INF-y increased in the probiotic group	[91]
Randomized, controlled, double-blind study with 159 newborns	LGG	10^10^ CFU/day for the first 6 months of life	Eczema and asthma	The estimated cumulative incidence of eczema and asthma was lower in the probiotic group at 2 years of age	[92]
Prospective, double-blind, placebo-controlled, randomized study with 40 children	*B. longum* BB536, *Bifidobacterium infantis* M-63, and *B. breve* M-16 V	Oral supplementation containing *B. longum* BB536 (3 × 10^9^ CFU), *B. infantis* M-63 (1 × 10^9^ CFU), and *B. breve* M-16 V (1 × 10^9^ CFU) as powder in a 3 mg sachet. Administered every day for 8 weeks	Seasonal allergic rhinitis and intermittent asthma	A significant improvement of symptoms and quality of life in the probiotic group	[67]
Double-blind, 2-arm, placebo-controlled study with 50 children with AD	*B. lactis* CECT 8145, *B. longum* CECT7347, and *L. casei* CECT 9104	10^9^ CFU/day of a mixture of the 3 probiotic strains	Atopic dermatitis	A reduction in IL-4, IL-5, and IL-13 and a decreased activity of Th2 in the probiotic group. The SCORAD index and use of corticosteroids were also reduced in the probiotic group	[93]
A multi-center, double-blind, placebo-controlled, randomized trial with 1099 very preterm infants	*B. infantis* BB-02, *Streptococcus thermophilus* TH-4, and *B. lactis* BB-12	A combination of *B. infantis* BB-02 (300 × 10^6^ CFU), *S. thermophilus* TH-4 (350 × 10^6^ CFU), and *B. lactis* BB-12 (350 × 10^6^). Total: 1 × 10^9^ CFU per 1.5 g in a powder once daily, until discharged from hospital or term-corrected age	Eczema, atopic sensitization, food allergy, and wheezing	There was no difference in eczema incidence between the two groups. Additionally, the incidence of atopic eczema, food allergy, wheezing, and atopic sensitization were similar in both groups	[52]

**Table 2 foods-10-00701-t002:** Use of probiotics to treat or prevent allergic diseases in adults and mother–infant pairs.

Type of Study	Probiotic	Dosage and Time of Exposure	Allergic Disease	Main Findings	Reference
Double-blind, randomized, placebo-controlled study with 159 mothers with allergic diseases and their infants	*Lactobacillus rhamnosus* GG (LGG)	1 × 10^10^ CFU daily for 2–4 weeks	Atopic eczema, asthma, and allergic rhinitis	The incidence of eczema in the probiotic group was halved. The concentrations of total immunoglobulin (Ig) E and positive reactions in skin-pick tests were similar in both groups	[55]
Randomized, double-blind, placebo-controlled trial with 44 adults	*Bifidobacterium longum* BB536	5 × 10^10^ CFU twice daily for 13 weeks	Japanese cedar pollinosis (JCPsis)	*B. longum* reduced severe symptoms and medication and decreased nasal blockage in rhinorrhea, and composite scores. Improvements in all symptoms	[58]
Randomized trial with newborns of 231 women with allergy	*Lactobacillus acidophilus* LAVRI-A1	3 × 10^9^ CFU/day for the first 6 months of life	Atopic dermatitis	Atopic dermatitis rates were similar in the probiotic and placebo groups. At 12 months, the rate of sensitization was significantly higher in the probiotic group	[85]
Double-blind, randomized, placebo-controlled study with 232 mothers	*L. reuteri* ATCC 55730	1 × 10^8^ CFU/day from gestational week 36 until delivery. Their babies then continued with the same product from birth until 12 months	Eczema	The cumulative incidence of eczema was similar in the two groups. The probiotic group had less IgE-associated, and skin prick test reactivity was also less common	[56]
Randomized, double-blind, placebo-controlled study with 2 parallel groups of 1223 pregnant women	LGG, *L. rhamnosus* LC705, *Bifidobacterium breve* Bb99, and *P. freudenreichii* spp. *shermanii* JS	Oral administration of LGG (5 × 10^9^ CFU)*, L. rhamnosus* LC705 (5 × 10^9^), *B. breve* Bb99 (2 × 10^8^), and *Propionibacterium freudenreichii* spp. *shermanii* JS (2 × 10^8^) twice daily for 2 to 4 weeks before delivery. Their infants received the same probiotics (+ prebiotics) once daily for the first 6 months	Food allergy, eczema, asthma, and allergic rhinitis	Administration of these probiotics significantly prevented eczema and atopic eczema	[94]
Placebo-controlled, double-blind study with 171 mother–infant pairs	LGG and *B. lactis* Bb12	10^10^ CFU/day of each probiotic from the first trimester of pregnancy to the end of exclusive breastfeeding	Atopy (eczema)	The concentration of TGF-b2 was higher in colostrum from dams supplemented with probiotics. This supplementation produced a protective effect against sensitization in infants at a high hereditary risk	[95]
Two-center, double-blind, randomized, placebo-controlled trial with 512 pregnant women and 474 infants	*L. rhamnosus* HN001 and *Bifidobacterium animalis* spp. *lactis* HN019	Two treatment groups: 6 × 10^9^ CFU/d *of L. rhamnosus* or 9 × 10^9^ CFU/d of *B. animalis* from 35 weeks of gestation until 6 months in pregnant women and from birth to 2 years in infants	Eczema and atopy	Supplementation with only *L. rhamnosus* reduced the cumulative prevalence of eczema, but not atopy, by 2 years	[96]
Double-blind, placebo-controlled, prospective study with 105 pregnant women	LGG	5 × 10^9^ CFU twice daily. Started 4–6 weeks before expected delivery until a postnatal period of 6 months	Atopic dermatitis	There was no significant difference between the probiotic group and placebo groups	[97]
Randomized, double-blind, placebo-controlled trial with 112 pregnant women	*Bifidobacterium bifidum* BGN4, *B. lactis* AD011, and *L. acidophilus* AD031	Mixture of probiotics (1.6 × 10^9^ CFU/day of each probiotic) for 4–8 weeks before delivery until 6 months after delivery	Eczema	The prevalence of eczema in the probiotic-added group was lower. The cumulative incidence of eczema during the first 12 months was reduced in the probiotic group. No difference in the serum total IgE level or sensitization to food allergens	[7]
Double-blind, randomized, placebo-controlled study with 156 pregnant women	*B. bifidum* W23 and *B. lactis* W52	1 × 10^9^ CFU/day of each strain during the last 6 weeks of pregnancy and postnatally for 12 months to their offspring	Eczema	The prevalence of eczema during the first 3 months of life was significantly lower in the probiotic group. The cumulative incidence was also lower in the probiotic group	[98]
Randomized, double-blind trial of children from 415 mothers	LGG, *B. animalis* ssp. *lactis* Bb-12, and *L. acidophilus* La-5	Milk contained 5 × 10^10^ CFU/day of *L. rhamnosus* and Bb-12. 5 × 10^9^ CFU of *L. acidophilus* for 4 months, from 36 weeks of gestation to 3 months postnatally	Atopic dermatitis and asthma	In the probiotic group, the cumulative incidence of atopic dermatitis was reduced, but there was no effect on sensitization	[89]
Double-blind, placebo-controlled study with 36 subjects with allergic rhinitis	*L. rhamnosus* GR-1 and *Bifidobacterium adolescentis* 7007-05	Added *L. rhamnosus* at 4% and *B. adolescentis* at 10% of final milk volume	Rhinitis	Serum IL-10 and IL-12 levels were increased in the probiotic group at the end of the grass pollen season. Additionally, the serum TFG-β levels were higher during the ragweed season	[99]
Randomized controlled trial with 250 pregnant women	LGG	1.8 × 10^10^ CFU/day from 36 weeks of gestation until delivery	Eczema	Prenatal probiotic treatment was only associated with decreased breast milk soluble CD14 and IgA levels	[100]
Randomized, double-blind, placebo-controlled study	*Lactobacillus plantarum* CJLP133	0.5 × 10^10^ CFU twice a day for 12 weeks	Atopic dermatitis	SCORAD scores were lower with probiotic administration. Additionally, the total eosinophil counts were significantly lower, and the logarithmic IFN-c and IL-4 concentrations were decreased	[101]
Parallel, double-blind, placebo-controlled study of 241 mother–infant pairs	*L. rhamnosus* LPR, *B. longum* BL999, and *Lactobacillus paracasei* ST11	Combination of *L. rhamnosus* and *B. longum* or *L. paracasei* and *B. longum*. Daily dose for each probiotic (1 × 10^9^ CFU) for 2 months before delivery and during the first 2 months of breastfeeding. The infants were followed for 24 months	Eczema	The risk of eczema was reduced in infants of mothers receiving combinations of probiotics during the first 24 months	[102]
Prospective, double-blind, placebo-controlled study with 191 pregnant women	LGG	1 × 10^10^ CFU/day from the second trimester of pregnancy	Atopic diseases	No significant effects of prenatal and postnatal probiotic supplementation on sensitization, development of allergic diseases, and maternal IgE levels. The allergic symptoms improved in the probiotic group	[103]
Double-blind, parallel-group, placebo-controlled comparison with 49 AD patients	*L. acidophilus* L-92	20.7 mg/day for 8 weeks	Atopic dermatitis	This probiotic contributes to the suppression of Th-2-dominant inflammation and reduces atopic dermatitis symptoms in adults	[104]
Randomized controlled study with 415 pregnant women	LGG, *B. animalis* spp. *Lactis* Bb-12, and *L. acidophilus* La-5	Daily dose of LGG (5 × 10^10^ CFU), *B. animalis* spp. *Lactis* Bb-12 (5 × 10^10^ CFU), and *L. acidophilus* La-5 (5 × 10^9^ CFU) from 36 weeks of gestation to 3 months postnatal	Atopic dermatitis	Supplementation with a combination of these probiotics reduced the proportion of Th22 cells. However, the proportion of Tregs, Th1, Th2, and Th17 cells and the Th1/Th2 ratio in the offspring were not affected	[105]
Double-blind, placebo-controlled, parallel, randomized clinical study with 173 participants	*Lactobacillus gasseri* KS-13, *B. bifidum* G9-1, and *B. longum* MM-2	Each capsule (350 mg) contained *L. gasseri* KS-13 (1.2 billion CFU), *B. bifidum* G9-1 (0.15 billion), and *B. longum* MM-2 (0.15 billion)	Rhinoconjunctivitis-specific	An improvement in the rhinoconjunctivitis-specific quality of life during allergy season was seen in the probiotic group	[106]
Randomized, double-blind, placebo-controlled study with 22 AD subjects	*L. plantarum* IS-10506	10^10^ CFU/day for 12 weeks	Atopic dermatitis	Reduction in clinical symptoms in AD children and decreased SCORAD and levels of serum IgE, IL-4, and IL-17. The probiotic acted through the downregulation of the Th2-adaptive immune response	[107]

## Data Availability

No new data were created or analyzed in this study. Data sharing is not applicable to this article.

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
