# Peer review of "Probiotics as a Possible Strategy for the Prevention and Treatment of Allergies. A Narrative Review"

_foods, 2021, doi:10.3390/foods10040701_

Round 1

Reviewer 1 Report

Relevent data were presented in the study conducted by Lopez-Santamarina et al. However, some aspects need to be addressed by the authors.

What kind of review did the authors conducted? This needs to be clear expressed in the manuscript and in the title.

In the abstract it should be pointed out the main results of the present study. Statements like this should be avoided: “We found interesting results related to the treatment…”

More detailed information regarding probiotics should be provided in the Introduction.

Why this work is relevant? Which gaps the present work will fill in the present scientific knowledge? This needs to be pointed out in the Introduction.

How did the studies included in the present review were selected? What were the inclusion criteria? A methodological section should be added to the manuscript.

Further references should be discussed by the authors, like:

Raposo, A., Pérez, E., Tinoco de Faria, C., & Carrascosa, C. (2017). Allergen management as a key issue in food safety. Food Safety and Protection; Ravishankar Rai, V., Bai, JA, Eds, 195-242.

Homayouni Rad, A., Aghebati Maleki, L., Samadi Kafil, H., & Abbasi, A. (2021). Postbiotics: A novel strategy in food allergy treatment. Critical reviews in food science and nutrition61(3), 492-499.

Lunjani, N., Satitsuksanoa, P., Lukasik, Z., Sokolowska, M., Eiwegger, T., & O'Mahony, L. (2018). Recent developments and highlights in mechanisms of allergic diseases: microbiome. Allergy73(12), 2314-2327.

Patil, S. V., Mohite, B. V., & Patil, V. S. (2021). Probiotics for Allergic Airway Infection and Inflammations. In Probiotic Research in Therapeutics (pp. 295-313). Springer, Singapore.

Author Response

Rebuttal Letter MS foods-1132363

Many thanks for the careful revision that been done to our article.  All the comments and suggestion highlighted by reviewers have been considered and corrected or incorporated in the revised version of the manuscript. We thank to the Reviewers because their comments allowed to strongly improve our manuscript.

With respect to the comments from the Reviewer 1:

With respect to the comments about “Relevant data were presented in the study conducted by Lopez-Santamarina et al.”

We greatly appreciate the constructive comments from the Reviewer.

With respect to the comments about “What kind of review did the authors conducted? This needs to be clear expressed in the manuscript and in the title”.

Thank you very much. Obviously, this is a narrative review and according to the suggestions from the reviewer it was specified in both title and main text.

With respect to the comments about “In the abstract it should be pointed out the main results of the present study. Statements like this should be avoided: “We found interesting results related to the treatment…”.

Thank you very much. The Reviewer are right and consequently, the cited phrase was deleted from the revised version of the manuscript.

With respect to the comments about “More detailed information regarding probiotics should be provided in the Introduction”.

Thank you for your comment. According to the suggestion from the Reviewer, in the Introduction section it was added more information about probiotics, in concrete:

“Probiotic” means “for life” and are defined as “Live microorganisms that when being administered in appropriate doses, confer a benefit to the health of the host” [5]. The use of probiotics to treat or prevent specific diseases is a branch of microbiology known as "therapeutic microbiology" [17-20]. Probiotics exert multiple health effects such as im-munomodulatory agents and activators of host defense pathways, reducing morbidity [21]. In the context of allergic disease, the probiotics microbiome is essential for develop-ment of host immune responses. Effective probiotics must be resistant to bile salts, gastric enzymes, low pH and do not cause mucosal inflammation or infection [22].”

 With respect to the comments about “Why this work is relevant? Which gaps the present work will fill in the present scientific knowledge? This needs to be pointed out in the Introduction.”

Thank you for your comments. The main relevance of this work is to provide an unified source of information where the latest evidence published in the scientific literature on the usefulness of probiotics for the prevention and symptomatic treatment of allergic diseases can be consulted. There are other previously published reviews on the use of probiotics in childhood allergies and in some specific adult pathologies, such as allergic rhinitis. However, we believe that a more extensive work, such as the present one, is useful for the scientific community. To reinforce this idea, in the revised version of the manuscript it was added the following paragraph in the Introduction section:

“The relevance of this work is to provide a broaden knowledge about new alternatives for the treatment of these diseases in both children and adults, reducing the symptoms asso-ciated with their allergies, and preventing the occurrence of allergen-caused acute adverse episodes.”

With respect to the comments about “How did the studies included in the present review were selected? What were the inclusion criteria? A methodological section should be added to the manuscript.”

Please note that as it was clarified in response to your previous comment, this is a narrative review and not a systematic review. In any case, for the realization of this review it was made a previous find of related articles and book chapters included in scientific finders, with special emphasis of most recently published articles. In order to clarify it, in the revised version of the manuscript, it was included a new subheading describing Materials and methods, cited as:

“2. Materials and methods

A narrative literature search was conducted up to 10 February 2021 for all available literature in the following databases: Web of science, PubMed and Scopus, without any re-striction. A combination of the following search terms was applied: ‘probiotics’ and ‘allergic diseases’; “gut microbiota” and “allergic diseases”. The search terms were adjusted to the specific databases and consisted of a combination of thesaurus terms and free-text search.

The following data on study characteristics were extracted from the included records: author and year of publication, type of study, probiotics used, dosage and time of administration, type of allergy and conclusions of each study.”

With respect to the comments about “Further references should be discussed by the authors, like:”

Raposo, A., Pérez, E., Tinoco de Faria, C., & Carrascosa, C. (2017). Allergen management as a key issue in food safety. Food Safety and Protection; Ravishankar Rai, V., Bai, JA, Eds, 195-242.

Homayouni Rad, A., Aghebati Maleki, L., Samadi Kafil, H., & Abbasi, A. (2021). Postbiotics: A novel strategy in food allergy treatment. Critical reviews in food science and nutrition61(3), 492-499

.Lunjani, N., Satitsuksanoa, P., Lukasik, Z., Sokolowska, M., Eiwegger, T., & O'Mahony, L. (2018). Recent developments and highlights in mechanisms of allergic diseases: microbiome. Allergy73(12), 2314-2327.

Patil, S. V., Mohite, B. V., & Patil, V. S. (2021). Probiotics for Allergic Airway Infection and Inflammations. In Probiotic Research in Therapeutics (pp. 295-313). Springer, Singapore.

Thank you for your comment. According to the suggestions from the Reviewer, all the cited documents were incorporated to the references list and discussed in the main text.

Reviewer 2 Report

The paper “Are probiotics an alternative strategy for the prevention and  treatment of allergies? A review” by Lopez-Santamarina et al., reports several aspects related to probiotic bacteria to verify their application as alternative to chemicals in the treatment of allergies and also to analyse their role in allergy prevention. In general, a review article should have some novelty, while this manuscript looks more descriptive rather than informative. From this perspective, I suggest the authors to provide a more critical summary of the current evidences. I am not sure the authors have a great experience in this field to write a review on this topic. From self-citations, I can only find a paper on Lactobacillus bulgaricus in murine allergic asthma; for this reason, a critical revision of the presented data would be expected.

  • I personally do not agree with question marks in title. They highlight the doubts about an assumption and put the reader into a doubtful concern. In my opinion, the Title should report a statement, a conclusion, as is it seems a “main hypothesis” around which the work is developed.
  • In addition to potential and beneficial aspects, the abstract of review article should also reports pitfalls, limitations and future prospects (intended to overcome the negative aspects).
  • L20-21 to be moved after “symptom alleviation” at L16.
  • L25-29. Since all information refer to [1], it is useless to use [twice].
  • It should be explained why the number of allergic people is going to quadruplicate in future. Is it enough to merge with what reported at L39-46? Or it is based on other processes?
  • L53-54. Confused. Please rewrite the sentence.
  • Recent directives indicate the number of probiotic per food portion.
  • In general, the mechanisms of probiotics to reduce allergy effects are not clearly reported. They are somewhat summarized in Table 2, but it is too schematic for a review article. The mechanisms should be grouped per bacterial species.
  • Table 2, extremely long, several information which could be better discussed in the text.
  • A paragraph on “Future prospects” reporting the future directions of research is mandatory for a review article.

Author Response

Rebuttal Letter MS foods-1132363

Many thanks for the careful revision that been done to our article.  All the comments and suggestion highlighted by reviewers have been considered and corrected or incorporated in the revised version of the manuscript. We thank to the Reviewers because their comments allowed to strongly improve our manuscript.

With respect to the comments from the Reviewer 2:

With respect to the comments about “The paper “Are probiotics an alternative strategy for the prevention and treatment of allergies? A review” by Lopez-Santamarina et al., reports several aspects related to probiotic bacteria to verify their application as alternative to chemicals in the treatment of allergies and also to analyse their role in allergy prevention. In general, a review article should have some novelty, while this manuscript looks more descriptive rather than informative. From this perspective, I suggest the authors to provide a more critical summary of the current evidences. I am not sure the authors have a great experience in this field to write a review on this topic. From self-citations, I can only find a paper on Lactobacillus bulgaricus in murine allergic asthma; for this reason, a critical revision of the presented data would be expected.”

Thank you for your comments. Please note that as it was clarified in the revived version of the manuscript, this is a narrative review, thus, it is reasonable that manuscript be more descriptive than informative, because it is not our intention to discuss or refute the data published by other authors, but to compile them in order to make them more easily accessible to the reader. The reviewer is right about that we do not have a great deal of specific experience in the treatment of allergic pathologies. However, we do believe we have a reasonable understanding of the human intestinal microbiota, its modulation by external agents, and the prevention of chronic noncommunicable diseases through the intestinal microbiota, including the use of probiotics. At the present time, we are carrying a research project about using probiotics to alleviate food-related allergic symptoms.

In any case, in the revised version of the manuscript, we tried to provide a more in deep description of important issues related to this use of probiotics, including some personal assessments, without criticizing or refuting previously published data.

With respect to the comments about “I personally do not agree with question marks in title. They highlight the doubts about an assumption and put the reader into a doubtful concern. In my opinion, the Title should report a statement, a conclusion, as is it seems a “main hypothesis” around which the work is developed.”

Thank you for your comments. It was not our intention to introduce doubt in the reader about the usefulness of probiotics in the symptomatic treatment of allergies; it was only a way of expressing the debate that gave rise to the article and the research contained in it. However, we think that the Reviewer is right and in the revised version of the manuscript the title was changed to “Probiotics as an alternative strategy for the prevention and treatment of allergies. A narrative review.”

With respect to the comments about “In addition to potential and beneficial aspects, the abstract of review article should also reports pitfalls, limitations and future prospects (intended to overcome the negative aspects).”

Thank you for your comments. According to the suggestion from the Reviewer, new information about reports pitfalls, limitations and future prospects was added to the Introduction section. In concrete:

“However, several risks can be associated with probiotic therapy in the clinical field, in vulnerable target groups such as pregnant women, infants, or immunodeficient subjects [23]. Medical studies have shown that, although uncommon, the use of probiotics may have some side effects as the spread of inappropriate resistance genes in intestinal microbial populations, virulence factors in probiotic microbial strains, translocation to tissues and blood, an inflammatory response, and infections such as abdominal abscesses, bacteremia, fungemia, peritonitis, urological infections, infective endocarditis, meningitis, pneumonia, rheumatic vascular diseases [27,110]. Commercial probiotics have also been found to contain live microorganisms not listed on their labels, which can be fatal to an individual. In fact, a premature infant has been reported to have died from fulminant gas-trointestinal mucormycotic due to fungal contamination of a probiotic supplement used. Therefore, it is also important to use probiotics with some caution [111, 112].

In addition, it should be noted that some of the studies collected have not shown clear effects of probiotics in treating allergies [52,85,88,98]. The works that did not found beneficial effects on humans included use of Lactobacillus [83,96] and mixes with Bifidobacterium [88]; or Streptococcus [52], and were applied to preterm infants [52] child [52,85] and pregnant women [98]. The only link between these studies is that in all cases probiotics were used for the treatment or prevention of atopic dermatitis, so it seems that in this pathology the use of probiotics is not as effective as in other allergic diseases..”

Comments about the negative results and the risk of certain probiotics species were also cited in the abstract section.

With respect to the comments about “L20-21 to be moved after “symptom alleviation” at L16.”

According to the suggestion from the Reviewer, the comments about L20-21 were moved accordingly.

With respect to the comments about “L25-29. Since all information refer to [1], it is useless to use [twice].”

Thank you for your comment. According to the suggestion from the Reviewer, we deleted the first time that reference [1] was cited in the text.

With respect to the comments about It should be explained why the number of allergic people is going to quadruplicate in future. Is it enough to merge with what reported at L39-46? Or it is based on other processes?

Thank you for your comment. According to the suggestion from the Reviewer, new information about the causes of the allergic diseases increase was added to the Introduction section. In concrete:

“Additionally, it was largely documented that living in rural areas at child age consti-tutes a factor conveying a protective effect against allergic disease, such as atopy, allergic rhinitis, or asthma [10,11], related to environmental-derived factors such as contact with bacterial endotoxins [12], contact with animals [13] or intake of raw or low processed milk [13,14]. In recent decades, migration from traditional farming to urban environment, in-crease in processed food intake, lack of contact with animals and excessive hygiene has been related with the increase in allergic diseases incidence [15,16].”

With respect to the comments about “L53-54. Confused. Please rewrite the sentence.”

Thank you for your comment. According to the suggestion from the Reviewer, the phrase in lines 53-54 was rewritten in the revised version of the manuscript. It was changed to “The intestinal microbiota is a key factor in the development of an adequate immune response because contact of gut bacterial antigens with immune system associated with the intestine represents an important component of the human immune system [17].”

With respect to the comments about “Recent directives indicate the number of probiotic per food portion.”

Thank you for your comment. According to the suggestion from the Reviewer, it was specified that according to actual consensus, a number of viable cells of 1x109 CFU/day in food and food supplements is considered the minimum number to be effective.

With respect to the comments about “In general, the mechanisms of probiotics to reduce allergy effects are not clearly reported. They are somewhat summarized in Table 2, but it is too schematic for a review article. The mechanisms should be grouped per bacterial species.”

Thank you for your comment. In the revised version of the manuscript, we included more information about probiotic mechanisms to reduce allergy. Please note that in most cases, the probiotics used in the trials are mixtures, and the concrete mechanisms for reduction for allergic symptoms are not fully understood and are not cited by authors. Considering that as was incorporated in the revised of the manuscript not only influence in the probiotic action the type and number of cells, but also the metabolites and byproducts (postbiotic), the citation of all mechanisms would force us to make assumptions that might not be correct.

With respect to the comments about “Table 2, extremely long, several information which could be better discussed in the text”.

Thank you for your comment. Because Table 1 was deleted from the revised version of the manuscript due to the suggestions from other reviewers, and the fact that you are right in that the Table is too long, in the revised version we slitted the Table 2 in different tables, including results related to infants and adults.

Additionally, more information was included in the main text, according to the suggestions from the Reviewer.

With respect to the comments about “A paragraph on “Future prospects” reporting the future directions of research is mandatory for a review article.”

According to the suggestion from the Reviewer, a paragraph called “Limitations, conclusions and future trends” was included in the revised version of the manuscript.

“5. Limitations, conclusions and future trends

Some of the studies have limitations, for example, the reduced sample size, the randomization process, and the small number of bacterial groups used. In some cases, the probiotic dosage employed are below the minimum consensus (109 UFC/day) for appropriate use probiotics. So, the results obtained is these works are of doubtful validity.

Although generally the use of most probiotics is safe, the use of some bacterial opportunistic pathogens, such as Enterococcus or Escherichia is controversial and should be avoid in especially vulnerable collectives. Previous studies have shown that, although uncommon, the use of probiotics may have some side effects.

This review has collected different published studies that show the beneficial effects of using probiotics to treat allergic diseases. Even considering that probiotics do not cure these diseases, their administration can improve the quality of life of patients by reducing the duration of illness and intensity of disease symptoms. An alternative to the administration of probiotics in supplement form would be to consume them in functional foods, as these would be more acceptable to consumers than pharmacological presentations and easier to consume. Although many studies have been conducted, more are needed to observe the true effects of probiotic consumption on allergic diseases, but this review gives an idea that more benefits are likely to be found in the administration of probiotics to treat allergies and possibly be administered alongside other treatments, thus reducing the morbidity people living with allergies. Additionally, the efficacy of probiotics is dependent also from factors such as composition and activity of the subject's intestinal microbiota and the derived metabolites and byproducts synthesized by the probiotic species. Thus, prior knowledge of the state of the subject's intestinal microbiota prior to the administration of the probiotic agent would be advisable to improve its effectiveness.”

Reviewer 3 Report

Title: Try to avoid question market in the title. After accomplished a proper review you may have a more clear and prompt title without pushing positively to much  of the use of probiotic products as you have to stay neutral and objective.

2. Abstract: I miss the word "global". The allergies are increasing, but is it a global problem? Any difference between cities and rural areas?

Avoid: "We found interesting results.." instead: "We found results.." How to classify "quality of life"?

3. Introduction: When mentioning "hygiene hypothesis" I miss the reference by Strachan BMJ (1989) and also the publications about the revised hygiene hypothesis: Blomfield et al (2006)Clinical and experimental Allergy 36. Just to use one solely reference nr 5 is not enough. You should empasize the difference between cities and rural environments. Also when it comes to reference nr 6-there are more references from the same area of research that should be cited.

Line 142-143: There is an unclear sentence: "In addition fungi and bacteria make up approximately 1% of the species in our gut microbiota."??

Line 153: Take away "beneficial"

Line 159: What is "Overall good health"? You may refer to WHO:s definition on health!

Please make a larger distinction between the animal studies and those clinical studies performed on human beings in all discussions!! I would recommend you to skip the animal studies and develop just the clinic studies performed on human beings.

Reference nr 37 is written in Spanish and is published by Nestlé?

Line 202- 223 is interesting, for instance the references regarding microbiota and diversities. Please develop that. In the Figure 1 "genetic predisposition" is mentioned. It may be the microbiome?

Line 248 there is a reference regarding Enterococcus, reference nr 50. Please also discuss the risks with plasmids?? Is there any plamids in the Enterococcus spp?

As several studies show no effect it should be highlighted. 

You may withdraw all studies performed on mice and develop the area of research performed on human beings?

Conclusion: You cannot write the line 290-292: "Consequently, probiotics are a good alternative strategy to prevent or to reduce allergic diseases symptoms, which affect many adults and especially children and young people." Delete!

What about ingestion of fermented foods? 

Please try to find some more references to each citation. For instance reference nr 1, 5 and 6 are solely cited. If this is going to be a review the citations are very important. I would also recommend you not to be too positive in your attitude as research must be objectively performed. Try to give more factual arguments on probiotic products! 

Wish you good luck with the revise process and future publishing as this area of research is very interesting and important!

Author Response

Rebuttal Letter MS foods-1132363

Many thanks for the careful revision that been done to our article.  All the comments and suggestion highlighted by reviewers have been considered and corrected or incorporated in the revised version of the manuscript. We thank to the Reviewers because their comments allowed to strongly improve our manuscript.

With respect to the comments from the Reviewer 3:

With respect to the comments about “Title: Try to avoid question market in the title. After accomplished a proper review you may have a more clear and prompt title without pushing positively to much of the use of probiotic products as you have to stay neutral and objective.

Thank you for your comment. According to the suggestion from the Reviewer, the question markers was deleted from the title and a neutral title was used, thus changing the title to “Probiotics as a possible strategy for the prevention and treatment of allergies. A narrative review. In the whole text, a more neutral position was adopted.

 With respect to the comments about “Abstract: I miss the word "global". The allergies are increasing, but is it a global problem? Any difference between cities and rural areas?

Thank you for your comment. In fact, allergies increase is nowadays a global concern, and is a more pressing problem for people living in urban environments than in rural environments. Thus, in the revised version of the manuscript, the first phrase of the abstract section was changed to “Allergies are an increasing global public health concern, especially for children and for people living in urban environments”.

With respect to the comments about “Avoid: "We found interesting results.." instead: "We found results.." How to classify "quality of life"?”

Thank you for your comment. The Reviewer are right and consequently, the cited phrase was deleted from the revised version of the manuscript. With respect to the term “quality of life”, we classify the “quality of life” as the as the fact that a patient does not suffer symptoms of his/her illness, and therefore such illness does not condition his/her life. In fact, the translation from Spanish to English may be confusing to the reader, and thus, in the revised version of the manuscript we changed from “improving quality of life for allergy sufferers” to “reducing morbidity for allergy sufferers”.

With respect to the comments about “3. Introduction: When mentioning "hygiene hypothesis" I miss the reference by Strachan BMJ (1989) and also the publications about the revised hygiene hypothesis: Blomfield et al (2006) Clinical and experimental Allergy 36. Just to use one solely reference nr 5 is not enough. You should empasize the difference between cities and rural environments. Also when it comes to reference nr 6-there are more references from the same area of research that should be cited.”

Thank you for your comment. According to the suggestion from the Reviewer, the 2 references cited by the Reviever (Strachan (1989) and Blomfield et al (2006)) were incorporated to the revised version of the manuscript, was well other 5 articles related to “hygiene hypothesis”.

The following paragraph were incorporated in the Introduction section: “Additionally, it was largely documented that living in rural areas at child age constitutes a factor conveying a protective effect against allergic disease, such as atopy, allergic rhinitis, or asthma [10,11], related to environmental-derived factors such as contact with bacterial endotoxins [12], contact with animals [13] or intake of raw or low processed milk [13,14]. In recent decades, migration from traditional farming to urban environment, in-crease in processed food intake, lack of contact with animals and excessive hygiene has been related with the increase in allergic diseases incidence [15,16].”.

Additionally, other 3 new articles were incorporated to reinforce the idea of probiotics as “therapeutic microbiology”.

With respect to the comments about “Line 142-143: There is an unclear sentence: "In addition fungi and bacteria make up approximately 1% of the species in our gut microbiota."??”

Thank you for your comment. In fact, this was a mistake and instead of “bacteria”, it should be read “protozoa”. The mistake was addressed in the revised version of the manuscript.

With respect to the comments about “Line 153: Take away "beneficial"”

Thank you for your comment. According to the suggestions from the Reviewer, the term “beneficial” was corrected and changed to “various”.

With respect to the comments about “Line 159: What is "Overall good health"? You may refer to WHO:s definition on health!”

Thank you for your comment. According to the suggestions from the Reviewer, the term “overall good health” was changed to “improve health”.

With respect to the comments about “Please make a larger distinction between the animal studies and those clinical studies performed on human beings in all discussions!! I would recommend you skip the animal studies and develop just the clinic studies performed on human beings.

Thank you for your comment. Please note that animal studies were separated from human trials in the fact that they are allocated in different Tables. Although animal tests obviously do not have the same relevance as human tests and are usually a preliminary phase to them, we believe it is important to mention them because there are some strains that, although not yet tested in humans, have shown beneficial activities in animals, and therefore may be valuable information for other research groups. According to your suggestion, we deleted in the revised version of the manuscript the Table 1, but maintaining a little reference to animal trials in the main text.

In any case, you are right with the fact that animal studies are of les relevance that human ones, and it was clarified in the revised version of the manuscript. In this order, it was incorporated in the main text:

“Obviously, experimental animals (mice in all cases), differ widely from humans in key aspect such as gut microbiota composition, immune function, or metabolism [45]. Thus, extrapolating the results obtained from animal models to humans may not be valid and always should be considered as preliminary or limited results.”

With respect to the comments about “Reference nr 37 is written in Spanish and is published by Nestlé?”

Thank you for your comment. A Spanish version of this article is available and with free access, so we used the Spanish version for a better understanding. In the revised version, the cited reference was changes to its English version.

With respect to the comments about “Line 202- 223 is interesting, for instance the references regarding microbiota and diversities. Please develop that. In the Figure 1 "genetic predisposition" is mentioned. It may be the microbiome?”

Thank you for your comment. In fact, microbiome is a factor that can influence the effectiveness of a probiotic in a single patient. According to your suggestion, in the revised version of the manuscript, it was added the following paragraph:

“The impact of probiotics on humans is highly-variable between individuals. In example, a recent work found that individuals with healthier gut microbiota composition responded in a better way than other individuals after administration of probiotics [61]. This result is logical, because probiotics action depends on several factors including colonization resistance, acid and short chain fatty acid production, competitive exclusion of pathogens or bacteriocin production can influence probiotic persistence [62,63]. Zmora et al [64] demonstrated that personalized gut mucosal persistence resistance to commercial probiotics is associated with unique host and microbiome features. In example, colonization of a probiotic can be affected by an under-representation of specific carbohydrate-utilization genes in the gut, thus preventing complex polysaccharides metabolization by the probiotic bacteria [62]. Thus, the efficacy of probiotics is dependent on both microbial species or strains and its derived metabolites and byproducts (commonly named as postbiotics), as well as the number of probiotic cells and type of probiotic carrier [27].”

With respect to the comments about “As several studies show no effect it should be highlighted.”

Thank you for your comment. The fact that several studies did not show any beneficial effects from probiotic use was reinforced in the text. In the revised version of the manuscript, it was included the following paragraph:

“In addition, it should be noted that some of the studies collected have not shown clear effects of probiotics in treating allergies [52,85,88,98]. The works that did not found beneficial effects on humans included use of Lactobacillus [83,96] and mixes with Bifidobacterium [88]; or Streptococcus [52] and were applied to preterm infants [52] child [52,85] and pregnant women [98]. The only link between these studies is that in all cases probiotics were used for the treatment or prevention of atopic dermatitis, so it seems that in this pathology the use of probiotics is not as effective as in other allergic diseases.”

With respect to the comments about “Line 248 there is a reference regarding Enterococcus, reference nr 50. Please also discuss the risks with plasmids?? Is there any plamids in the Enterococcus spp?”

According to the Reviewer´s suggestions, we included some information about the risk of using Enterococcus as probiotics. In the revised version of the manuscript, it was included the following paragraph:

“Use of Enterococcus as probiotics presents some intrinsic risks that makes them poorly suited for use as probiotics, especially E. faecium and E. faecalis species. As opportunistic pathogens, both enterococci possess an array of virulence factors and antibiotic resistance determinants that render them controversial. Even in avirulent enterococci, when they reach intestinal tract, they can interact with endogenous enterococci and other commensals that could lead to bi-directional genetic exchange [65,66]”

With respect to the comments about “As several studies show no effect it should be highlighted.”

Thank you for your comment. Please note that animal studies were separated from human trials in the fact that they are allocated in different Tables. Although animal tests obviously do not have the same relevance as human tests and are usually a preliminary phase to them, we believe it is important to mention them because there are some strains that, although not yet tested in humans, have shown beneficial activities in animals, and therefore may be valuable information for other research groups.

In any case, you are right with the fact that animal studies are of les relevance that human ones, and it was clarified in the revised version of the manuscript. In this order, it was incorporated in the main text:

“Obviously, experimental animals (mice in all cases), differ widely from humans in key aspect such as gut microbiota composition, immune function, or metabolism [45]. Thus, extrapolating the results obtained from animal models to humans may not be valid and always should be considered as preliminary or limited results.”

With respect to the comments about “You may withdraw all studies performed on mice and develop the are of research performed on human beings”

According to the suggestions from the Reviewer, the Table 1 and most refences to mice studies were deleted, maintain a little presence in the main text was well as clarifying the difference relevance of animal and human trials.

With respect to the comments about “Conclusion: You cannot write the line 290-292: "Consequently, probiotics are a good alternative strategy to prevent or to reduce allergic diseases symptoms, which affect many adults and especially children and young people." Delete!”

Thank you for your comment. According to the suggestions from the Reviewer, the cited phrase was deleted.

With respect to the comments about “What about ingestion of fermented foods?”

Thank you for your comment. In this review we have not advised the intake of probiotics through pharmaceutical presentations, we have only mentioned the species, doses and results obtained. Obviously, we consider better the administration through food than through pharmaceutical presentations (note that this article is intended to be published in Foods). In order to clarify it and according to the suggestions from the Reviewer, in the revised version of the manuscript we included a short paragraph in this sense:

“Probiotics can be intake by pharmaceutical preparations or by functional foods, generating wider acceptance among consumers, mainly by fermented foods. Because intake of functional foods shows better consumer acceptance than pharmacological presentations, there is potential for foods that are consumed regularly when they are converted to functional foods [49]. There are also studies showing that intake fermented foods, such as fermented yeast, galactooligosaccharides or fructooligosaccharides, can improve the symptoms of some allergies such as asthma or eczema by reducing oxidative stress and inflammation [113-115]. This suggests that the combined administration of probiotics and prebiotics could be useful for alleviating symptoms for allergic patients.”

With respect to the comments about “Please try to find some more references to each citation. For instance, reference nr 1, 5 and 6 are solely cited. If this is going to be a review the citations are very important. I would also recommend you not to be too positive in your attitude as research must be objectively performed. Try to give more factual arguments on probiotic products!”

Thank you for your comment. According to the suggestion form the Reviewer more references were added to 1,5 and 6. Other references were incorporated along the main text (in total 27). Nevertheless, please consider that in order to comply your recommendations of delete trials carried out in experimental animals’ results, some of the references cited in the original version of the manuscript were deleted from the revised version.

With respect to the too positive attitude, in the revised version a more neutral a more neutral tone. In this sense, a new paragraph including limitations and piftails of the manuscript were incorporated.

With respect to the comments about “Wish you good luck with the revise process and future publishing as this area of research is very interesting and important!”

Thank you for your constructive comments. We will try to make contributions in this area in the near future.

Round 2

Reviewer 1 Report

Now, the manuscript is suitable in its present form.

Author Response

We greatly appreciate the Reviewer participation in the review process that allowed a great improvement of the manuscript quality.

Reviewer 2 Report

All comments have been addressed modifying the text and/or providing appropriate rebuttals

Author Response

(The authors gave the same response as above.)

Reviewer 3 Report

Dear authors,

Thank you for considering all my suggestions for change.

Author Response

The response to Reviewer´s comments is placed in the attached pdf file
